# Chest CT Findings after 4 Months from the Onset of COVID-19 Pneumonia: A Case Series

**DOI:** 10.3390/diagnostics10110899

**Published:** 2020-11-03

**Authors:** Luigi Urciuoli, Elvira Guerriero

**Affiliations:** 1Department of Radiology, G. Criscuoli Hospital, Sant’Angelo dei Lombardi, 83100 Avellino, Italy; luigiurciuoli@yahoo.it; 2Department of Advanced Biomedical Sciences, University “Federico II”, Via S Pansini, 5, 80131 Naples, Italy

**Keywords:** COVID-19, chest CT, follow-up, 4 months

## Abstract

Coronavirus disease 2019 (COVID-19) is an infectious disease caused by the severe acute respiratory syndrome coronavirus 2 (SARS-CoV-2). Although the reference standard for SARS-CoV-2 diagnosis is real-time reverse transcription polymerase chain reaction (RT-PCR), computed tomography (CT) is recommended for both initial evaluation and follow-up. There is a growing body of published evidence about CT evolution during the course of COVID-19 pneumonia. Here, we report six confirmed cases of COVID-19 patients who underwent unenhanced chest CT on admission and after 4 months from the onset of symptoms. Chest-CT at first admission showed the typical CT features of COVID-19. Interestingly, the follow-up CT revealed the persistence of lung abnormalities in five cases even if all the patients were completely asymptomatic. Further studies are needed for a comprehensive understanding of the disease progression and the resulting late imaging modifications.

## 1. Introduction

COVID-19 (coronavirus disease 2019) is caused by severe acute respiratory syndrome coronavirus 2 (SARS-CoV-2) [1]. First identified in China in December 2019 [2], it has since become a pandemic.

Although the reference standard for SARS-CoV-2 diagnosis is real-time reverse transcription polymerase chain reaction (RT-PCR), computed tomography (CT) is recommended for both initial evaluation and follow-up [3]. CT findings of COVID-19 pneumonia vary with time and different temporal stages have been described [4].

Here, we present a series of six patients with a mild form of COVID-19 pneumonia, who underwent unenhanced chest CT at first assessment and after 4 months from the onset of symptoms. The repeated chest CT showed in five cases the persistence of lung abnormalities, although all the patients presented a complete clinical resolution of pneumonia.

## 2. Case Series

None of our patients were smokers or suffered from smoking-related lung diseases.

In all six cases, RT-PCR performed on the patients’ nasal swab samples was positive for SARS-CoV-2. The patients were placed in isolation and received symptomatic and antibiotic treatment with no need of oxygen therapy; all patients were discharged after 4 weeks when they became completely asymptomatic and when 2 consecutive respiratory samples tested negative for SARS-CoV-2. We repeated chest CT after 4 months from the onset of symptoms and about after 3 months from the hospital discharge. All procedures performed in the studies involving human participants were in accordance with the ethical standards of the institutional and/or national research committee and with the 1964 Helsinki Declaration and its later amendments or comparable ethical standards. Informed consent was obtained from all individual participants included in the study.

### 2.1. Case 1

A 45-year-old woman, previously in good health, presented to the emergency department (ED) with myalgia, asthenia, and a body temperature of 37.4 °C. Oxygen saturation (sO_2_) was 98% and all laboratory results were normal. 

On admission, an unenhanced chest CT showed multiple, rounded ground-glass opacities (GGOs) with peripheral distribution, especially in the right lower lobe (Figure 1a,b). 

A chest CT at 4 months from the onset of symptoms showed complete reabsorption of GGOs with no lung abnormalities (Figure 1c,d).

### 2.2. Case 2

A 65-year-old woman was admitted to the hospital with a fever (38 °C) and slight cough. Laboratory tests revealed a slightly decreased white blood cell count (WBC) 3.7 × 10^9^/L (4–11 × 10^9^/L) and a normal absolute lymphocyte count (ALC) 1.5 × 10^9^/L (0.9–2.9 × 10^9^/L). Procalcitonin-PCT 0.15 ng/mL (<0.5 ng/mL), C-reactive protein-CRP 6 mg/L (<10 mg/L), and sO_2_ (98%) were within normal limits. 

On admission, chest CT showed a diffuse “crazy paving pattern” peripherally located in the lower lobes (Figure 2a,b).

A chest CT at 4 months from the onset of symptoms showed the persistence of mixed pattern characterized by GGOs and fibrous streaks, bilaterally located. (Figure 2c,d).

### 2.3. Case 3

A 68-year-old woman with a 20-pack-year smoking history was admitted to hospital with a seven-day history of fever (38 °C), chest tightness, and muscle pain. Coarse breath sounds at the bases of both lungs were heard on auscultation. sO_2_ was 96%, and laboratory tests showed a normal WBC 10 × 10^9^/L (4–11 × 10^9^/L), ALC 1.3 × 10^9^/L (0.9–2.9 × 10^9^/L), CRP 8 mg/L (<10 mg/L), and PCT 0.4 ng/mL (<0.5 ng/mL).

On admission, chest CT showed a diffuse “crazy paving pattern” peripherally located in the upper and lower lobes (Figure 3a–c).

A CT at 4 months from the onset of symptoms revealed the bilateral persistence of a mixed pattern characterized by GGOs and fibrous streaks (Figure 3d–f).

### 2.4. Case 4

A 49-year-old previously healthy man presented to the ED with a five-day history of sore throat and fever (37.8 °C). The level of sO_2_ was 98%, and laboratory tests showed a normal WBC 6 × 10^9^/L (4–11 × 10^9^/L), ALC 2.5 × 10^9^/L (0.9–2.9 × 10^9^/L), CRP 5.5 mg/L (<10 mg/L), and PCT 0.4 ng/mL (<0.5 ng/mL). 

On admission, chest CT showed patchy areas of GGO with in upper lobes (Figure 4a,b), in the middle and left lower lobe (Figure 5a,b).

A CT at 4 months from the onset of symptoms showed the bilateral persistence of fibrotic stripes (Figure 4c,d and Figure 5c,d).

### 2.5. Case 5

A 66-year-old man was admitted to the hospital with a four-day history of dry cough, fever, and myalgia. He had a ten-year history of hypertension. At presentation, his temperature was 37.7 °C. sO_2_ was 97%, and laboratory tests showed a normal WBC 5 × 10^9^/L (4–11 × 10^9^/L), ALC 1.5 × 10^9^/L (0.9–2.9 × 10^9^/L), CRP 1.5 mg/L (<10 mg/L), and PCT 0.3 ng/mL (<0.5 ng/mL).

On admission, chest CT showed a “crazy paving pattern” peripherally located in the upper left lobe and in the lower lobes (Figure 6a,b).

A CT at 4 months from the onset of symptoms showed bilateral persistence of mixed pattern characterized by interlobular septal thickening and patchy GGOs (Figure 6c,d).

### 2.6. Case 6

A 64-year-old man presented with a five-day history of fever and recent onset dyspnea. The patient had no other comorbidities. Physical examination revealed tachypnea and a temperature of 39 °C. Coarse breath sounds were heard on auscultation at admission. The level of sO_2_ was 98%, and laboratory tests revealed neutrophilia 10 × 10^9^/L (1.7–7 × 10^9^/L), lymphopenia 0.7 × 10^9^/L (0.9–2.9 × 10^9^/L), an elevated CRP 112 mg/L (<10 mg/L), and elevated lactic acid dehydrogenase LDH 400 U/L (105–333 U/L).

On admission, chest CT showed bilateral extensive areas of GGO and consolidations, with prevalent peripheral distribution in the upper and lower lobes (Figure 7a–c). 

A CT at 4 months from the onset of symptoms showed the persistence of diffuse thickening of the interlobular septa, with fibrotic appearance (Figure 7d,e); air bubble sign with bronchiectasis (Figure 7f) was also recognizable. An air bubble sign consists of a small air-containing space that can result from a dilatation of a physiological space or from lung cystic changes or can be related to consolidation resorption.

## 3. Discussion

COVID-19 is an infectious disease caused by severe acute respiratory syndrome coronavirus 2 (SARS-CoV-2) [1], which is thought to have originated from a fish and wild animal market in Wuhan, the capital of China’s Hubei province, in December 2019 [2]. The WHO declared a global health emergency on 30 January 2020. 

COVID-19 typically presents with systemic and/or respiratory manifestations, especially fever, fatigue, and dry cough. Some patients have sputum production and dyspnea, and a few patients have symptoms such as headache, sore throat, muscle pain, nasal congestion, nausea and/or vomiting, and diarrhea. In severe cases, acute respiratory distress syndrome, septic shock, difficult to correct metabolic acidosis, and coagulation dysfunction develop rapidly [5].

The reference standard for the diagnosis of the infection is based on viral nucleic acid test, through RT-PCR on a throat swab [6].

Radiological examination plays an important role in the early detection and management of the infection. Chest-radiographs can be normal in early or mild disease. In this context, Chest CT is strongly recommended in suspected cases for both initial evaluation and follow-up [4,7]. A wide variety of CT findings have been reported in COVID-19 patients in different studies [8,9]. However, the main CT features of COVID-19 pneumonia are multifocal bilateral GGOs, typically with a peripheral and subpleural distribution. Pure GGO lesions can be an early sign of the infection. Isolated GGOs or a combination of GGOs and consolidative opacities are some of the most common CT findings, while pure consolidation is relatively less common or absent. In the majority of COVID-19 patients, multiple lobes are involved, especially the lower lobes [10,11].

The high-resolution CT (HRCT) features of COVID-19 pneumonia are non-specific and may be encountered in other lung infections caused by influenza virus and other coronaviruses such as SARS (severe acute respiratory syndrome) and MERS (middle east respiratory syndrome). In influenza-related pneumonia, chest CT shows a combination of small patchy ground glass and consolidative opacities with a subpleural and/or peri-bronchial distribution. In SARS, chest CT shows subpleural (unilateral or bilateral) GGOs associated with consolidation, especially in the lower lobes, and interlobular and intralobular septal thickening. In MERS, the common CT findings include diffuse bilateral subpleural GGOs associated with interlobular and intralobular septal thickening and pleural effusions.

Therefore, the CT features of these viral infections overlap, so it is mainly the current epidemic context that suggests COVID-19 as the cause of GGOs in patients with fever and respiratory symptoms [12].

There is a growing body of published evidence about CT evolution during the course of COVID-19 pneumonia. The data are still inhomogeneous, but it is reasonable to deduce that COVID-19 pneumonia CT features can change widely over time and that fibrotic stripes could be a hallmark of the late CT stage.

Wang et al. [13] described the CT findings according to 5 illness periods (0–5 days, 6–11 days, 12–17 days, 18–23 days, ≥24 days) and assigned a CT score dividing the lungs into six zones. CT scores and the number of involved lung zones increased rapidly, with a peak on stage “days 6–11”. The most common finding was GGO, which increased in the late stages, whereas consolidation was the second most observed finding in the first 11 days. 

Jin et al. [14] divided the most frequent imaging findings according to a classification in 5 stages: ultra-early, early (1–3 days from onset of symptoms), consolidation (7–14 days from onset of symp-toms), and the dissipation stage (2–3 weeks from onset of symptoms). GGOs were characteristics of the first two phases, consolidations of phases three and four, whereas the last stage showed a progressive increase of thickening of the interlobular septa and bronchial walls

Pan et al. [15] classified CT findings according to 4 temporal stages. GGO was recognizable in all the disease stages; crazy paving in the first three stages; consolidation was the most common finding in stage 3. In the second stage, the pneumonia rapidly worsened, with diffuse bilateral multilobar distribution. After day 14, imaging improvement was observed in 75% of patients. 

Lei et al. [16] observed that the patients with COVID-19 pneumonia had a typical transition from early stage to advanced stage and advanced stage to dissipating stage. The manifestations of single or multiple GGOs were observed in the early stage, higher density consolidations were presented in the advanced stage, and ground-glass opacities and consolidations were absorbed in the dissipating stage. In their study, patients with fibrosis in follow CT were older, with a longer length of stay (LOS), and a higher rate of Intensive Care Unit (ICU) admission than that of those who without fibrosis. 

Our six confirmed COVID-19 cases presented with mild symptoms such as asthenia, myalgia, cough, and fever. Chest CT, performed at first assessment, revealed the presence of the typical CT features of COVID-19 pneumonia: focal rounded pure GGOs or GGOs with smooth septal thickening located in both the costal and mediastinal subpleural peripheral parenchyma. The patients underwent a chest CT after 3 months from the onset of symptoms and about after 2 months from the hospital discharge. Interestingly, although no patients presented any respiratory complaints, only 1 showed a complete resolution of lung abnormalities. In particular, in 2 cases chest CT showed the persistence of a mixed pattern characterized by GGOs and fibrous streaks; in 1 one case fibrotic stripes were detectable; in 1 case bilateral persistence of mixed pattern characterized by interlobular septal thickening and patchy GGOs was observed; in 1 case chest CT showed a fibrotic pattern, characterized by diffuse thickening of the interlobular septa and air bubble sign with bronchiectasis. 

To our knowledge, this is one of the first work describing the CT features after 4 months from the onset of symptoms. Although limited by the small size sample, our case series demonstrates that the dissipation stage, described in the recent literature, could last for months after the clinical resolution of the disease. As of yet, we do not know if some risk factors, such as old age and comorbidities could predict the persistence of lung abnormalities.

Further studies are needed for a comprehensive understanding of the disease progression and of the underlying pathophysiology of this infective process. 

In conclusion, radiologists should be aware of the late modifications of image findings in the course of the disease in order to help patient management and implement proper treatment.

## Figures and Tables

**Figure 1 diagnostics-10-00899-f001:**
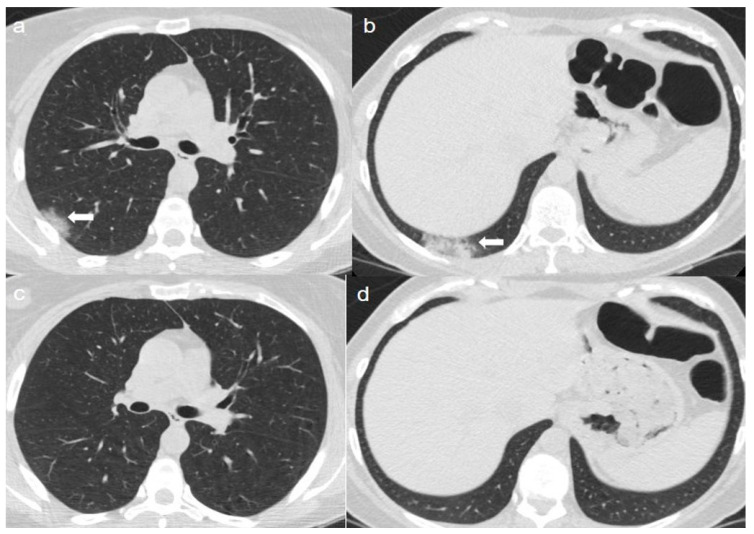
(**a**,**b**) CT shows areas of ground-glass opacities (GGO) with peripheral distribution in right lower lobe (arrows). (**c**,**d**) CT after 4 months from the onset of symptoms shows complete reabsorption of GGOs with no lung abnormalities.

**Figure 2 diagnostics-10-00899-f002:**
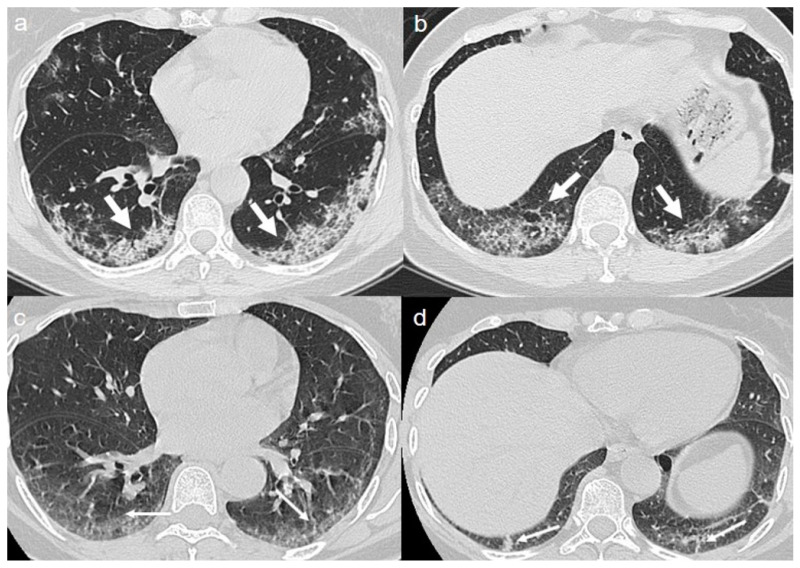
(**a**,**b**) CT shows a “crazy paving pattern” peripherally located in lower lobes (arrows). (**c**,**d**) CT after 4 months from the onset of symptoms shows bilateral persistence of mixed pattern characterized by GGOs (thin arrows in **c**) and fibrous streaks (thin arrows in **d**).

**Figure 3 diagnostics-10-00899-f003:**
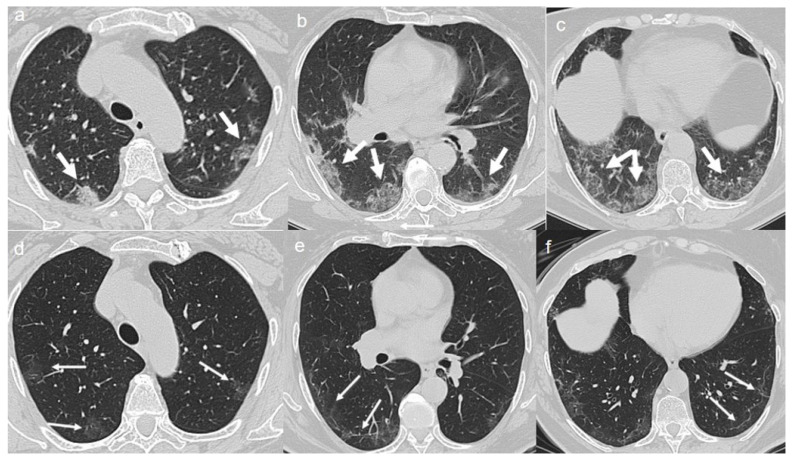
(**a**–**c**) CT shows a “crazy paving pattern” peripherally located in upper and lower lobes (arrows). (**d**–**f**) CT after 4 months from the onset of symptoms shows bilateral persistence of mixed pattern of GGOs (thin arrows in **d**) and fibrous streaks (thin arrows in **e**,**f**).

**Figure 4 diagnostics-10-00899-f004:**
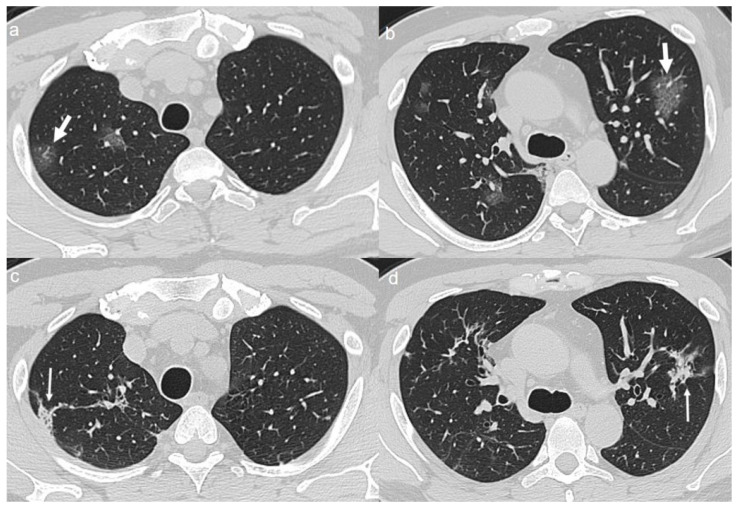
(**a**,**b**) CT shows patchy areas of GGO with in the upper lobes (arrows). (**c**,**d**) CT after 4 months from the onset of symptoms shows bilateral persistence of fibrotic stripes in upper lobes (thin arrows).

**Figure 5 diagnostics-10-00899-f005:**
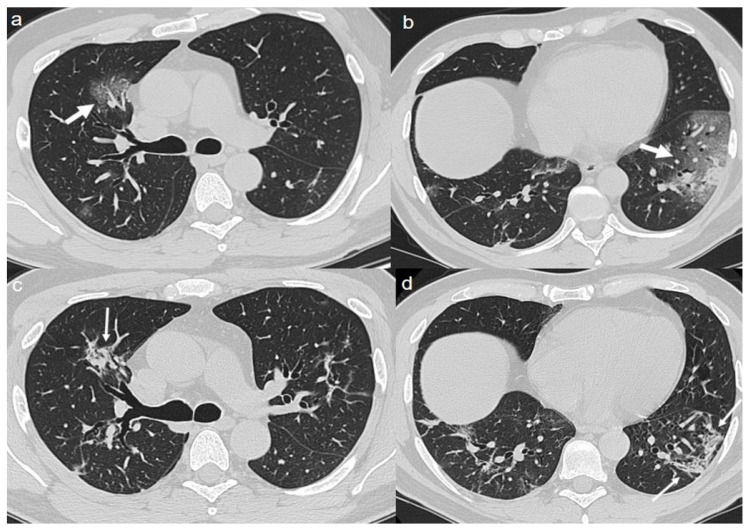
(**a**,**b**) CT shows patchy areas of GGO in the middle lobe (arrow in **a**) and in the left lower lobe (arrow in **b**). (**c**,**d**) CT after 4 months from the onset of symptoms shows bilateral persistence of fibrotic stripes in middle lobe (thin arrow in **c**) and in left lower lobe (thin arrows in **d**).

**Figure 6 diagnostics-10-00899-f006:**
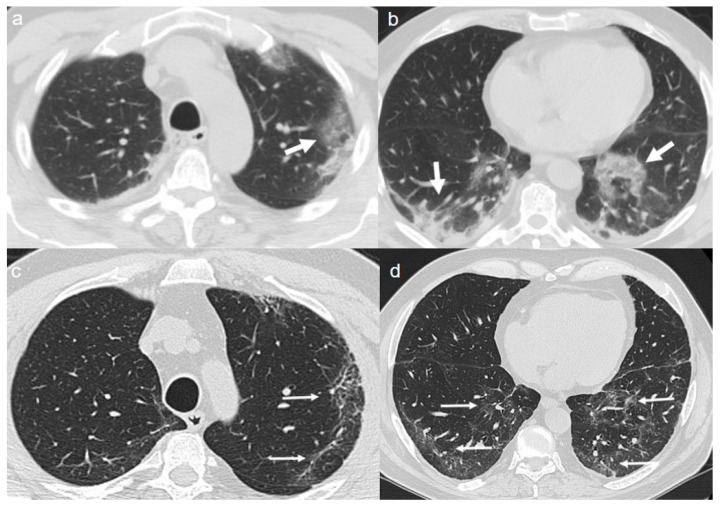
(**a**,**b**) CT shows a “crazy paving pattern” peripherally located in the upper left lobe (arrow in **a**) and in the lower lobes (arrows in **b**). (**c**,**d**) CT after 4 months from the onset of symptoms shows persistence of mixed pattern characterized by interlobular septal thickening (thin arrows in **c**) and patchy GGOs (thin arrows in **d**).

**Figure 7 diagnostics-10-00899-f007:**
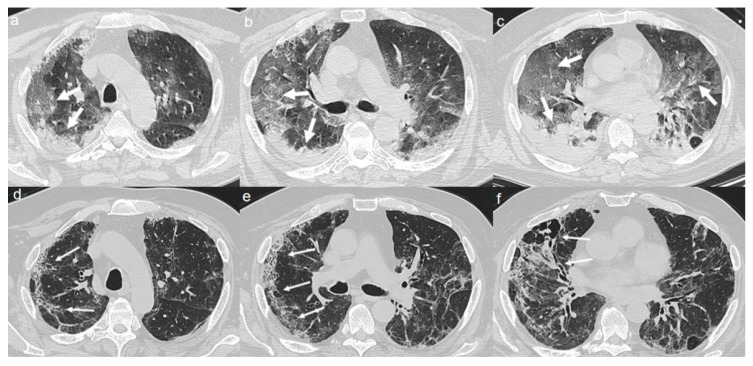
(**a**–**c**) CT shows bilateral extensive areas of GGO and consolidations, with prevalent peripheral distribution, in upper and lower lobes (arrows). (**d**–**f**) Follow up CT 4 months after admission shows persistence of diffuse thickening of the interlobular septa, with fibrotic appearance (thin arrows in **d**,**e**); air bubble sign with bronchiectasis (thin arrows in **f**) is recognizable.

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
