# Peer review of "Chest CT Findings after 4 Months from the Onset of COVID-19 Pneumonia: A Case Series"

_diagnostics, 2020, doi:10.3390/diagnostics10110899_

Round 1

Reviewer 1 Report

The case report provides interesting data, including follow-up CT conducted months after the onset of symptoms. However, it still needs a revision to be acceptable for this journal.

The quality of images a, b of figure 5 is low, comprared with the images c,d. Why is that?

Paragraphs 7-10 in the discussion section are poorly organized and redundant.

Author Response

RESPONSE TO REVIEWER 1 COMMENTS:

Point 1: The quality of images a, b of figure 5 is low, comprared with the images c,d. Why is that?

Response 1: We believe that the quality of all images in the manuscript is the same.

Point 2: Paragraphs 7-10 in the discussion section are poorly organized and redundant.

Response 2: We have summarized Paragraphs 7-10 in the discussion section. You can find it in red in the text (lines 199-206, pages 8-9).

Reviewer 2 Report

Thank you for your manuscript describing this CT Case series of COVID-19 patients that included results 4-months after initial diagnosis.

I have no particular qualms with the manuscript except that it would be very helpful to reference (from the literature) cases with other respiratory infections (e.g. influenza, other coronaviridae ...). While the comparison of 0 versus 4 months is useful, what distinguishes these image patterns from other infections? Also, I expect that some pathology would be apparent with cigarette smokers?

Author Response

RESPONSE TO REVIEWER 2 COMMENTS

Point 1:I have no particular qualms with the manuscript except that it would be very helpful to reference (from the literature) cases with other respiratory infections (e.g. influenza, other coronaviridae ...). While the comparison of 0 versus 4 months is useful, what distinguishes these image patterns from other infections?

Response 1: We added the image patterns of other respiratory infections (e.g influenza, SARS, MERS..). You can find it in red in the text (lines 207-218, page 9; lines 318-319, page 11)

Point 2: Also, I expect that some pathology would be apparent with cigarette smokers?

Response 2: None of our patients were smokers.We added this important information in the text in red (line 61, page 2)

Indeed, if the follow-up patterns had also been correlated with smoking, we would have found them also at the first assessment Chest-CT.